# Robust two-stage influenza prediction model considering regular and irregular trends

Taichi Murayama[1], Nobuyuki Shimizu[2], Sumio Fujita[2], Shoko Wakamiya[1], Eiji Aramaki[1] *

**1** Nara Institute of Science and Technology (NAIST), Ikoma-city, Japan, **2** Yahoo Japan Corporation, Tokyo, Japan

* aramaki@is.naist.jp

**Data Availability Statement:** All google trends data (google queries) are available from the google Trends site (https://trends.google.co.jp/) All US influenza surveillance reports are available from the FLUView (http://gis.cdc.gov/grasp/fluview/

## Abstract

Influenza causes numerous deaths worldwide every year. Predicting the number of influenza patients is an important task for medical institutions. Two types of data regarding influenza-like illnesses (ILIs) are often used for flu prediction: (1) historical data and (2) user generated content (UGC) data on the web such as search queries and tweets. Historical data have an advantage against the normal state but show disadvantages against irregular phenomena. In contrast, UGC data are advantageous for irregular phenomena. So far, no effective model providing the benefits of both types of data has been devised. This study proposes a novel model, designated the two-stage model, which combines both historical and UGC data. The basic idea is, first, basic regular trends are estimated using the historical data-based model, and then, irregular trends are predicted by the UGC data-based model. Our approach is practically useful because we can train models separately. Thus, if a UGC provider changes the service, our model could produce better performance because the first part of the model is still stable. Experiments on the US and Japan datasets demonstrated the basic feasibility of the proposed approach. In the dropout (pseudo-noise) test that assumes a UGC service would change, the proposed method also showed robustness against outliers. The proposed model is suitable for prediction of seasonal flu.

## Introduction

Seasonal influenza epidemics, representing severe infectious diseases, are characterized by the widespread incidence of various symptoms such as a sudden onset of fever, cough, headache, and muscle and joint pain. The World Health Organization (WHO) reported that 3–5 million cases of severe illness occur worldwide each year because of seasonal influenza, leading to approximately 290,000—650,000 deaths annually [1]. Seasonal influenza also affects the economic productivity because of employee absence and unexpected increases in hospital work loads [2]. Prediction of influenza outbreaks is therefore crucially important to support real-time decision-making related to the management of hospital resources with a rapid response.

fluportaldashboard.html) All Japan influenza surveillance reports are available from the NIID (https://www.niid.go.jp/niid/ja/idwr.html).

**Funding:** This study was supported in part by JSPS KAKENHI Grant Number JP19K20279, Health and Labor Sciences Research Grant Number H30-shinkougyousei-shitei-004, and Yahoo! Japan. The funder provided support in the form of salaries for authors (Dr Nobuyuki Shimizu and Mr Sumio Fujita), but did not have any additional role in the study design, data collection and analysis, decision to publish, or preparation of the manuscript. The specific roles of these authors are articulated in the 'author contributions' section.

**Competing interests:** This study was supported in part by JSPS KAKENHI Grant Number JP19K20279, Health and Labor Sciences Research Grant Number H30-shinkougyousei-shitei-004, and Yahoo! Japan. The funder provided support in the form of salaries for authors (Dr Nobuyuki Shimizu and Mr Sumio Fujita), but did not have any additional role in the study design, data collection and analysis, decision to publish, or preparation of the manuscript. This does not alter our adherence to PLOS ONE policies on sharing data and materials.

To predict influenza incidence, influenza-like illness (ILI) data of two types have been widely applied: (1) historical time series data, which mostly involve the previous year's data, and (2) online user generated content (UGC) data.

## Historical data

Historical time series data, called historical data, have a feature that potentially involves crucially important information such as marked seasonality. Therefore, approaches based on historical data have been reported [3–5], and various models [6–8] have been proposed. Kane et al. [9] compared the effectiveness of flu prediction of an autoregressive integrated moving average (ARIMA) model and a regression model based on random forest. Nasserie T., et al. [10] used disease models such as the IDEA model to project influenza peaks and epidemic final sizes. Some researches [11–13] used long short-term memory (LSTM), a neural network model, to verify its effectiveness in flu prediction. Yang Wan. et al. [14] showed that the application of statistical filtering methods to epidemiological models makes reliable influenza prediction possible by comparing the performance of six state-of-the-art filter methods such as particle filters. These researches improved the prediction performance and showed the effectiveness of data intensive prediction techniques of flu epidemics. Although numerous efforts have been made, these approaches suffer from one limitation; they are insufficient to discriminate against various unexpected sudden movements.

## UGC data

Another line of flu prediction research typically utilizes UGC text data or search query logs to seek signals of epidemic-related activities from crowds of users. To capture irregular movements detected by these signals, most studies have employed UGC on the Web [15] including search queries [3, 16, 17], microblogs [18–20], and access logs to Web pages such as Wikipedia [16] for flu prediction. Although a comparison of these resources has been presented in earlier reports [16, 21, 22], it still requires active discussions regarding which resources are the most useful for predicting flu epidemics. Signorini, A., et al. [22] examined the volume in posts including keywords related to influenza in the Twitter stream and showed the usefulness of twitter data for tracking flu epidemics. In addition, some have studies tackled the aspects such as which keywords on Google are useful [23] and when and where the model based on Twitter works well [20]. Various posts and search queries by users in UGC are important as signals to identify the beginning of epidemics and are used for the prediction of various epidemics [24] apart from influenza. Currently, Google Flu Trends (GFT) [17] is one of the most representative systems using UGC, which was designed to estimate the current ILI rate using Google search terms related to ILI. A benefit of using UGC data is that they can quickly detect the signals from online activities of a crowd of users. Nevertheless, UGC-based ILI prediction has some limitations: UGC exhibits difficulty in capturing crucially important regular trends such as long term annual movements in the time series of historical flu data, which might help the prediction of flu outbreaks [25].

In conclusion, these materials have their own advantages and disadvantages. They mostly correspond to two phenomena of epidemic management: (1) interpreting the intrinsic time series (regular trend) and (2) interpreting a sudden unexpected epidemic (irregular trend). Several studies [3, 18, 26] described the model of predicting the influenza volume using historical data and UGC data simultaneously and showed the effectiveness of the combination of each resource simultaneously. Instead of simultaneously, this study separately employs the two resources; the historical data are applied for regular trends, and the UGC data, for irregular trends (Fig 1). Our approach is practically useful because we can train the models separately;

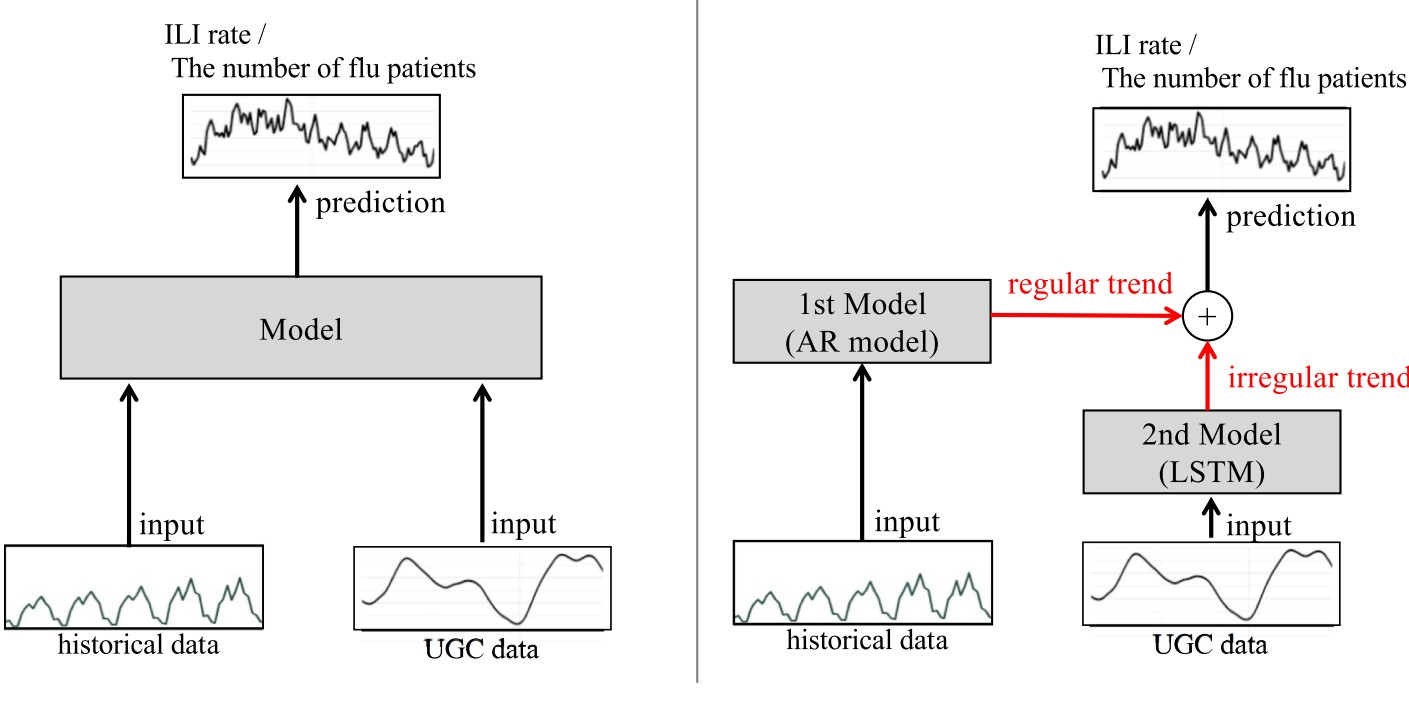

**Fig 1. Overview of the previous research (Left) and two-stage prediction model (Right).** Given historical data and UGC data, the existing approaches predict the influenza volume using historical and UGC simultaneously. In contrast, the Two-stage prediction model comprising the **1st model** and the **2nd model** can predict regular trends such as the periodicity from the historical data in the **1st model** and irregular trends such as sudden epidemics from the UGC data in the **2nd model**. Finally, the Two-stage model combines the regular and irregular trends from these models and predicts ILI activity such as the ILI rate in the US and the number of flu patients in Japan after a week.

in case a UGC provider changes its service, our model would only have to train the UGC model. Our model is also robust against noise and outliers in data because the model is divided into two parts: regular prediction using historical data and irregular trend prediction using UGC data. We validated our hypothesis regarding the robustness of the proposed method throughout the experiment in "Robustness" of the Results Section.

## Materials and methods

### Material

**US historical ILI data (CDC data).** In the US, the Centers for Disease Control and Prevention (CDC) provides weekly influenza surveillance reports called FluView. Information on outpatient visits to health care providers for ILI is collected through the US Outpatient Influenza-like Illness Surveillance Network (ILINet). The national percentage of patient visits to healthcare providers for ILI reported each week is calculated by combining state-specific data weighted by state population. Because of the time required for data aggregation and percentage calculation, the CDC reports have a delay of about a week. We use the ILI rate data from the 2010/40th week to the 2018/39th week as the US's historical ILI data.

**Japanese historical ILI data (NIID data).** Similar to the CDC, the National Institute of Infectious Diseases (NIID) reports data on the number of patients with ILI symptoms every week in Japan through the Infectious Disease Weekly Report (IDWR). These reports have a delay of approximately a week because of the time necessary for aggregating clinical

information, similar to CDC reports. However, the ILI patient number is more generic than the ILI rate in Japan, unlike the US. Therefore, we use the number of ILI patients from the 2009/40th week to the 2018/39th week as Japanese historical ILI data.

**UGC data (Google trends).**    Google Trend (GT) data has been used as UGC data. We randomly selected 20 query words from flu-related queries used in [3] as predictors and obtained the approximate weekly search volume data of these words from the 2012/40th week to the 2018/39th week. The time series of GT data were normalized to have a zero mean and a standard deviation of one.

Because flu-related words used in the US differ from those in Japan, translating English words into Japanese is insufficient to select the appropriate queries. For example, the abbreviation of influenza "flu" is translated into "I-N-FU-LU-E-N-ZA" and is not translated into the Japanese abbreviation of influenza "I-N-FU-LU". As an orthographical variant, the three categories of characters used by the Japanese three writing script are applied (kanji-script, hiragana-script, and katakana-script). Twenty queries were selected based on the correlation coefficients between the word frequency in Twitter and the number of patients with ILI symptoms. From GT, we obtained the approximate search volume weekly data of 20 words from the 2011/40th to the 2018/39th week. Like the US data, we normalized GT data such that the time series of GT data had a zero mean and a standard deviation of one. Our collection method complied with the terms and conditions for these data provision sites.

## Method: Two-stage model

**Basic idea.**    Our methodology can accommodate data of two types: historical data and UGC data, which have different benefits and shortcomings. To exploit the benefits of each kind, we proposed a novel method that divides the prediction process into two stages—the 1st model and the 2nd model—where each model uses different types of data.

1. **1st model for regular trends**: First, an autoregressive model is used to learn and predict the regular movements from only the historical data. We designate this model as the 1st model, which makes the basis of the succeeding model. The model predicts the future ILI rate/patient number from the historical data.

2. **2nd model for irregular trends**: This model is designed to predict sudden outbreaks. To identify sudden outbreaks, the model learns and predicts the difference between the ground truth and the predicted values by the 1st model using GTs data, in a different training term from the 1st model training term.

We designated the combination of the 1st model and 2nd model as the **Two-stage model**. Thus, the final prediction values of the Two-stage Model are calculated as the sum of the output value of the 1st model and that of the 2nd model. As each model is trained to predict a regular trend value and a deviated value from this, no coefficient weights are required to adjust them. Fig 2 shows the example of the prediction by the Two-stage model: the 1st model first predicts future ILI rate/patient number as regular trends from historical data (blue point), and then, the 2nd model predicts the differences between the output of the 1st model and the actual value (red dotted arrow), as irregular trends. The Two-stage model outputs the sum of the output of the 1st and 2nd model.

## Implementation

An autoregressive model (AR model) [27] is used for the 1st model. A LSTM model [11–13] is employed for the 2nd model. The Two-stage model calculates the sum of the output value of the 1st model and that of the 2nd model without any coefficient weights, as follows:

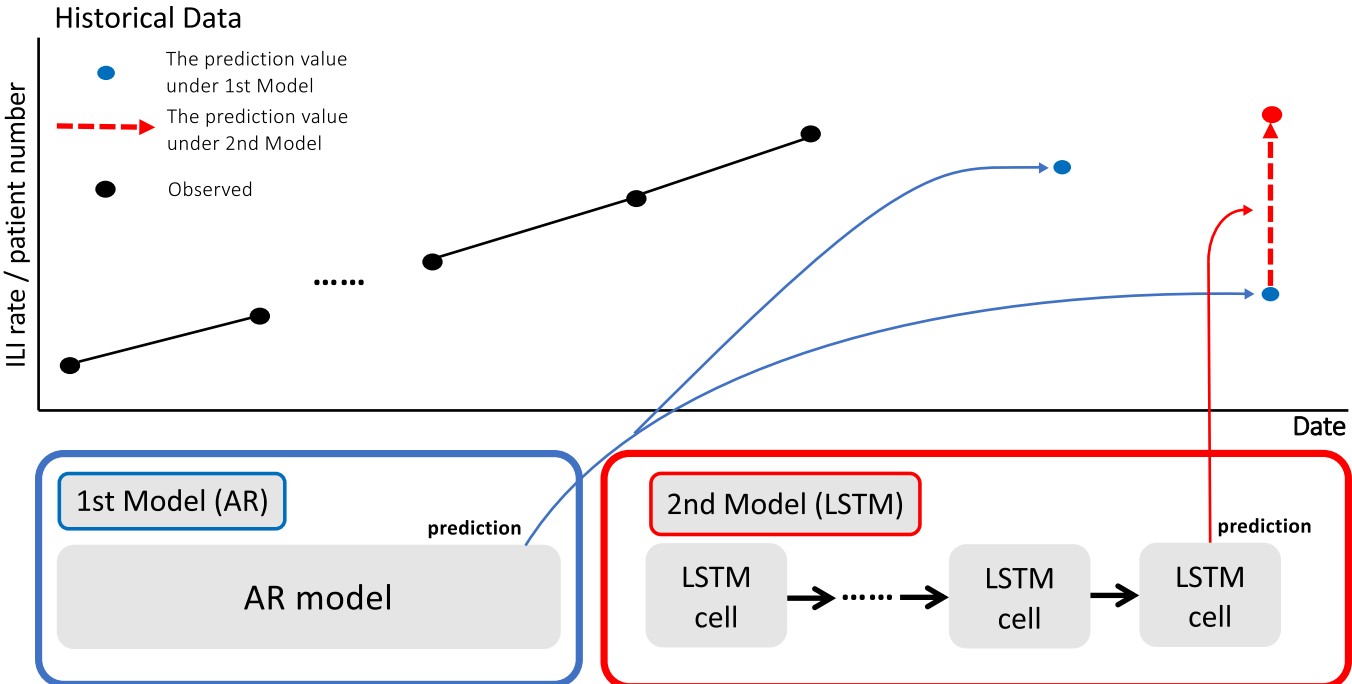

**Fig 2. Outline of the two-stage model: Black points in the upper graph indicate the observed values in each week.** A Two-stage model produces a one-week ahead flu prediction by a combination of the 1st and 2nd models. The 1st model, the input of which is the time series of historical data, predicts future ILI rate/ patient number as regular trends. The 1st model predicts the number of patients of this week and one-week ahead, given the historical data until the last week. The outputs of the 1st model are shown as blue points in the upper graph. Then, the 2nd model predicts irregular trends, which are differences between the predicted values and the actual values. The input for the 2nd model is time series data of the weekly total search frequencies of a set of predefined queries. The output of the 2nd model is shown as a dotted red arrow in the upper graph. The output of the Two-stage model is indicated by a red point, which is the simple sum of the outputs of the 1st and 2nd models.

**Autoregressive model (AR model).** The AR model is a type of random process, which is commonly used in time-series analyses. This model calculates each value of a time series using its own previous values. Given a time series, AR(p) has the following form.

$$y_t = c + \sum_{i=1}^{p} \varphi_i y_{t-i} + \varepsilon_t, \qquad (1)$$

Therein, $p$ represents the number of most recent values before a time $t$ which predicts the value of $t$: $y_t$. In addition, $\varphi_i$ stands for a parameter of the model, $c$ is a constant, and $\varepsilon_t$ denotes white noise.

Our paper uses the AR model as the 1st model. The input data for the AR model is the total weekly ILI rate/number data. It is aggregated by medical institutions in each area. We selected AR(26), which uses 26 weeks (a half year) of ILI data as predictors, and attempted to capture the seasonality of half a year. Although AR(52) would be the best to capture year-long seasonality as a year accounts for 52 weeks, it practically requires more data in training. Thus, it is difficult to perform a sufficient number of test verifications.

**Long Short-Term Memory (LSTM).** LSTM, which is a successful method for NLP applications [28, 29], processes a sequence of input and target pairs $[(x_1, y_1), \ldots, (x_n, y_n)]$. For each pair, it takes a new input $x_i$ and produces an estimate for a target $y_i$ using earlier inputs $x_{i-t}$, $\ldots, x_i$. An LSTM cell works as a memory to manage information according to the decisions specified by the input **I**, output **Y**, and forget gates **F**. Each memory cell is implemented as

shown below.

$$I_t = \sigma(W_{x_i}x_t + W_{m_i}o_{t-1} + b_i), \qquad Y_t = \sigma(W_{x_o}x_t + W_{m_o}o_{t-1} + b_o),$$
$$F_t = \sigma(W_{x_f}x_t + W_{m_f}o_{t-1} + b_f), \qquad \tilde{C}_t = (W_{x_c}x_t + W_{m_c}o_{t-1} + b_c), \tag{2}$$
$$C_t = F_t \odot C_{t-1} + I_t \odot \tanh(\tilde{C}_t), \qquad o_t = Y_t \odot \tanh(C_t)$$

Here, $W_x$ and $W_m$ are adaptive weights and $b$ is the intercept, initialized randomly in the range (0,1). In addition, $x_t$ and $o_{t-1}$ denote the current input and previous output vectors, respectively. The current cell state is denoted as $C_t$; $\sigma$ denotes the sigmoid function; and $\odot$ denotes the Hadamard product.

Our paper uses the LSTM as the 2nd model. The input data for LSTM are the weekly search query frequency data obtained through GT. We use 26 weeks of GT data before each prediction point as predictors. We predicted the differences between the ground truth and the predicted values by the 1st model. In Eq (2), $o_t$ denotes the predicted difference at time $t$. In Fig 2, we show the usage of LSTM mentioned above for our experiment; such usage is very similar to the method used in a previous study [12].

**The two-stage model.**    The Two-stage model outputs the sum of the output of the 1st model (AR model) and the 2nd model (LSTM). In our setting, the AR model first outputs the prediction of the ILI rate/patient number from historical data prior to more than two weeks, and then, the LSTM model outputs the prediction of sudden outbreaks, which is the difference between the ground truth and the output of the AR model. We predict the number of influenza patients at time $t+1$, as formulated as follows:

$$\begin{aligned}
\hat{y}_t^{1st} &= AR(y_{t-1}, y_{t-2}, ..., y_{t-26}), \\
\hat{y}_{t+1}^{1st} &= AR(\hat{y}_t^{1st}, y_{t-2}, y_{t-3}, ..., y_{t-25}), \\
\hat{d}_{t+1}^{2nd} &= LSTM(g_t, g_{t-1}, ..., g_{t-25}), \\
\hat{y}_{t+1} &= \hat{y}_{t+1}^{1st} + \hat{d}_{t+1}^{2nd}
\end{aligned} \tag{3}$$

The AR model, as the 1st model, predicts a regular trend, $\hat{y}_{t+1}^{1st}$ from the 26 weeks of the historical data $y$. The LSTM model, as the 2nd model, predicts the irregular trend $\hat{d}_{t+1}^{2nd}$ from 26 weeks of the GT data $g$. The output of the Two-stage model, $\hat{y}_{t+1}$, is calculated by adding $\hat{y}_{t+1}^{1st}$ and $\hat{d}_{t+1}^{2nd}$.

## Results & discussion

### Settings

The proposed model is evaluated by estimating ILI indexes (relative ratio in the US; the absolute patient numbers in Japan). The model produces a one-week forecast from a specific week using the ILI report delayed by one week in each country and GT data in the same week. Given $i$-th week $t_i$, the model estimates the ILI rate (in the US case) or the number of ILI patients (in Japan case) after a week.

We apply the AR(26) model as the 1st model using historical data for 26 weeks (a half year) as predictors. The LSTM model as the 2nd model uses GT data, the period of which is from a total of 26 weeks. It predicts the gap separating the ground truth and predicted values by the 1st model. The number of hidden layers in LSTM is one layer; the size of the hidden layer is selected from (5, 20, 32, 64, 128) in the validation period. We set 50 as the number of epochs. Fig 3 presents an example of prediction using the proposed model.

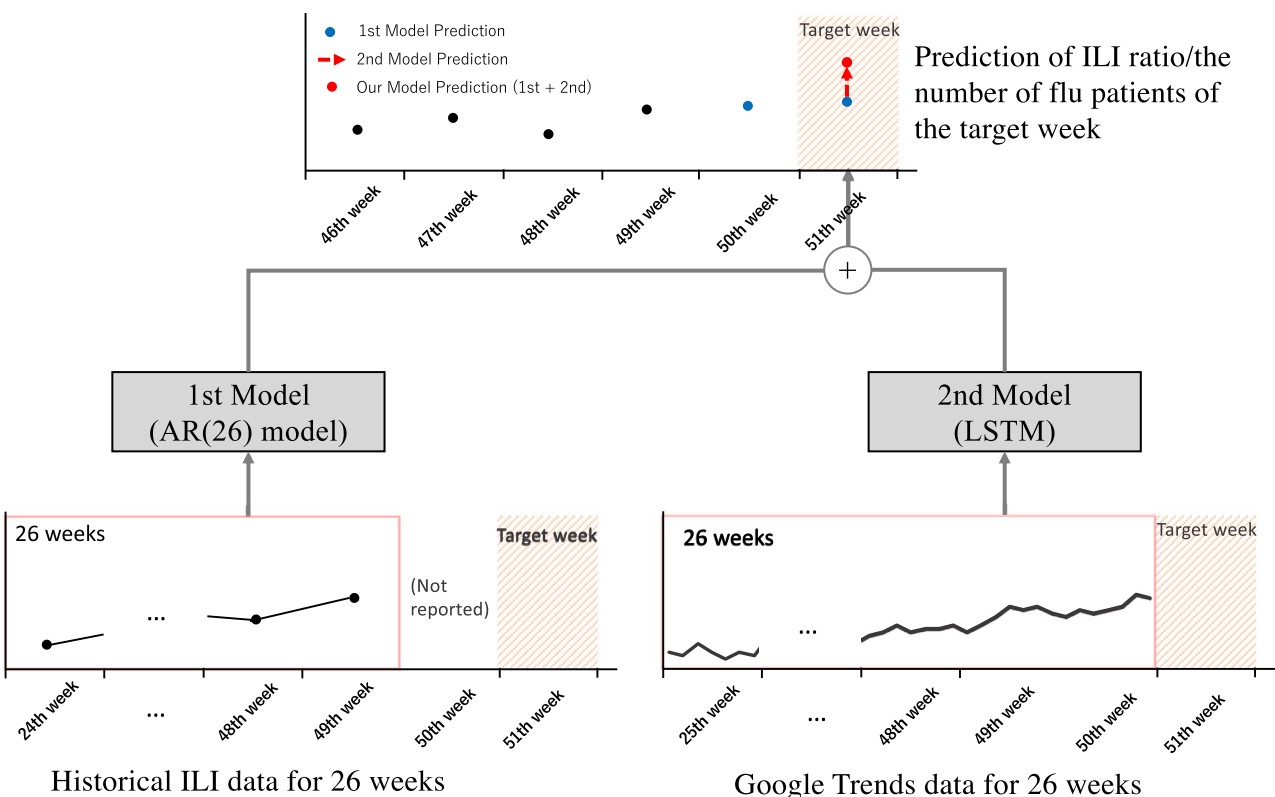

**Fig 3. Example of flu prediction using the two-stage model.** When estimating the ILI rate/ILI patient number for the target week of 51th week, the historical data for 26 weeks from 24th to 49th week are used for the 1st model because of an approximately one week delay before data are reported. For the 2nd model, GT data for 26 weeks from 25th to 50th week are available before the target week is used.

Table 1 presents an overview of training, validation, and test periods. For the experiments, we assessed the predictive performance using test data for one year. A year-long period is also set for the validation data. We set 104 weeks (two year-long) and 156 weeks (three year-long) for the 1st model training on historical data and the 2nd model training on GT data before the test period, respectively.

We also prepared datasets with outliers to evaluate the robustness of the proposed model. UGC data sometimes exhibit disadvantages against noise (outliers) due to malfunctions in crawling or those in services because of heavy traffic. These problems lead to confounded prediction and time-consuming preprocessing of the data. Robustness against noise and outliers is important. Thus, we investigated the robustness of the Two-stage model. ILI data is not processed because it is less susceptible to contain outliers. GT datasets with outliers were created

**Table 1. Overview of training, validation and test periods used for experiments.**

|  | Training | | Validation | Test |
|---|---|---|---|---|
|  | **1st model** | **2nd model** | | |
| US | 2010/40th–2012/39th | 2012/40th–2015/39th | 2015/40th–2016/39th | 2016/40th–2017/39th |
|  | 2011/40th–2013/39th | 2013/40th–2016/39th | 2016/40th–2017/39th | 2017/40th–2018/39th |
| Japan | 2009/40th–2011/39th | 2011/40th–2014/39th | 2014/40th–2015/39th | 2015/40th–2016/39th |
|  | 2010/40th–2012/39th | 2012/40th–2015/39th | 2015/40th–2016/39th | 2016/40th–2017/39th |
|  | 2011/40th–2013/39th | 2013/40th–2016/39th | 2016/40th–2017/39th | 2017/40th–2018/39th |

by randomly changing the frequency values of GT data in each query at each time span to 0.0 (the minimum value) or to the maximum value in each dataset without assuming any statistical distribution. Based on the above processing, three datasets with different ratios of outliers; 5.0%, 10.0%, and 15.0%, were created from the US GT data and the Japanese GT data, respectively. As described in the experimental settings, we conducted the same experiments on the prediction of the ILI rate and the number of ILI patients in the US and Japan from the 40th week of 2017 to the 39th week of 2018.

## Baseline methods

To evaluate the proposed method, we compared the proposed model with the two following models: the ARGO model and Random Forest Regression (RFR), which are well known for flu prediction [3, 7, 9].

**ARGO model.** The ARGO model [3], which is motivated by the hidden Markov model, performs autoregression on Google search data, in which Google search queries act as exogenous variables. The ARGO model is presented as follows.

$$y_t = \mu_y + \sum_{j=1}^{N} \alpha_j y_{t-j} + \sum_{i=1}^{K} \beta_i X_{i,t} + \varepsilon_t, \quad \varepsilon_t \overset{\sim}{} N(0, \sigma^2) \tag{4}$$

where $y_t$ represents the weighted ILI rate at time $t$, and $X_{i,t}$ is the result of term $i$ at time $t$ using GT. Here, $N$ stands for the number of weeks incorporated in ARGO to capture the seasonality in ILI activity. Then, as described also in an earlier report [3], we set $N = 52$ (weeks) to find parameters $\mu_y, \alpha_1 \ldots \alpha_{52}$, and $\beta_1 \ldots \beta_{20}$ that minimize the following.

$$\sum_t \left( y_t - \mu_y - \sum_{j=1}^{52} \alpha_j y_{t-j} + \sum_{i=1}^{20} \beta_i X_{i,t} \right)^2$$
$$+ \lambda_\alpha ||\alpha||_1 + \eta_\alpha ||\alpha||_2^2 + \lambda_\beta ||\beta||_1 + \eta_\beta ||\beta||_2^2 \tag{5}$$

Here, $\lambda_\alpha, \lambda_\beta, \eta_\alpha$, and $\eta_\beta$ are hyperparameters. The ARGO model employs L1 and L2 regularization to achieve automatic selection of the most relevant information separately for each data group. These hyperparameters are selected from $\lambda_\alpha, \lambda_\beta, \eta_\alpha = (0.001, 0.01, 0.1, 1.0)$ during the validation period.

**Random Forest Regression (RFR).** The random forest approach has been used for several public health studies such as the prediction of deer mouse population dynamics [30] along with influenza studies. This approach is a tree-based method that stratifies or segments the predictor space into several simple regions. It is frequently used to analyze variable importance.

## Evaluation metrics

Three evaluation metrics were used—the coefficient of determination $R^2$, mean absolute error (MAE), and mean absolute percent error (MAPE). $R^2$ is a measure of how well the predicted values conform to true values; the higher, the better. MAE is a measure of the average magnitude of differences between predicted values and true ones; the lower the better. Finally, MAPE is a measure of the average magnitude of the different ratio between predicted values

and true ones; the lower the better. These metrics are defined as presented below.

$$R^2 = 1 - \frac{\sum_{t=1}^{n}(F_t - A_t)^2}{\sum_{t=1}^{n}(A_t - \bar{A}_t)}, \qquad MAE = \frac{\sum_{t=1}^{n}|F_t - A_t|}{n} \qquad (6)$$

$$MAPE = \frac{1}{n}\sum_{t=1}^{n}\left|\frac{F_t - A_t}{A_t}\right| * 100\% \qquad (7)$$

where $n$ denotes the number of weeks. In addition, $A_t$ and $F_t$ respectively denote the true value and the predicted value at week $t$. $R^2$ is an important indicator among the three in terms of flu prediction because it shows how well a model fits and represents the accuracy of the prediction during epidemic seasons.

## Results

Table 2 and Fig 4 present the overall summary. In the US, the Two-stage model showed the best performance in some metrics of respective periods. However, no significant advantage was observed when compared to other models. In Japan, the Two-stage model exhibited the best performance in terms of all metrics for all periods.

## Model comparison

The Two-stage model achieved the best performance for several periods and metrics in the US and achieved the best performance among all metrics in Japan, demonstrating that the Two-stage model is better for flu prediction compared to the other two models. Especially, in MAPE, it outperformed the other compared models, indicating that suitable values were predicted.

The accuracy of ARGO in the US is close to that of the Two-stage model, but its accuracy in Japan is not as high as in MAE and MAPE. This indicates that ARGO highly depends on feature quantity. Compared to ARGO, the results from random forest seems to be a more stable result in every period. The predictive performance of the Two-stage model was higher than those of the ARGO and RFR models in many points of comparison.

**Table 2. Accuracy of the proposed model (two-stage model) and the models used for comparison (ARGO model (ARGO) and Random Forest Regression (RFR)).**

|  | Test period | Metrics | ARGO | RFR | Two-stage model |
|---|---|---|---|---|---|
| US | 2016/40th–2017/39th | $R^2$ | **0.964** | 0.950 | 0.935 |
|  |  | MAE | **0.171** | 0.200 | 0.216 |
|  |  | MAPE | 11.21 | 11.70 | **11.10** |
|  | 2017/40th–2018/39th | $R^2$ | 0.941 | 0.834 | **0.947** |
|  |  | MAE | 0.329 | 0.432 | **0.321** |
|  |  | MAPE | **13.92** | 16.68 | 15.09 |
| Japan | 2015/40th–2016/39th | $R^2$ | 0.895 | 0.840 | **0.914** |
|  |  | MAE | 8493.20 | 12375.92 | **6989.78** |
|  |  | MAPE | 62.97 | 179.22 | **34.47** |
|  | 2016/40th–2017/39th | $R^2$ | 0.528 | 0.739 | **0.745** |
|  |  | MAE | 14684.53 | 12143.82 | **8966.96** |
|  |  | MAPE | 110.59 | 58.13 | **31.48** |
|  | 2017/40th–2018/39th | $R^2$ | 0.837 | 0.817 | **0.863** |
|  |  | MAE | 16049.81 | 12923.58 | **10958.86** |
|  |  | MAPE | 989.64 | 54.68 | **31.66** |

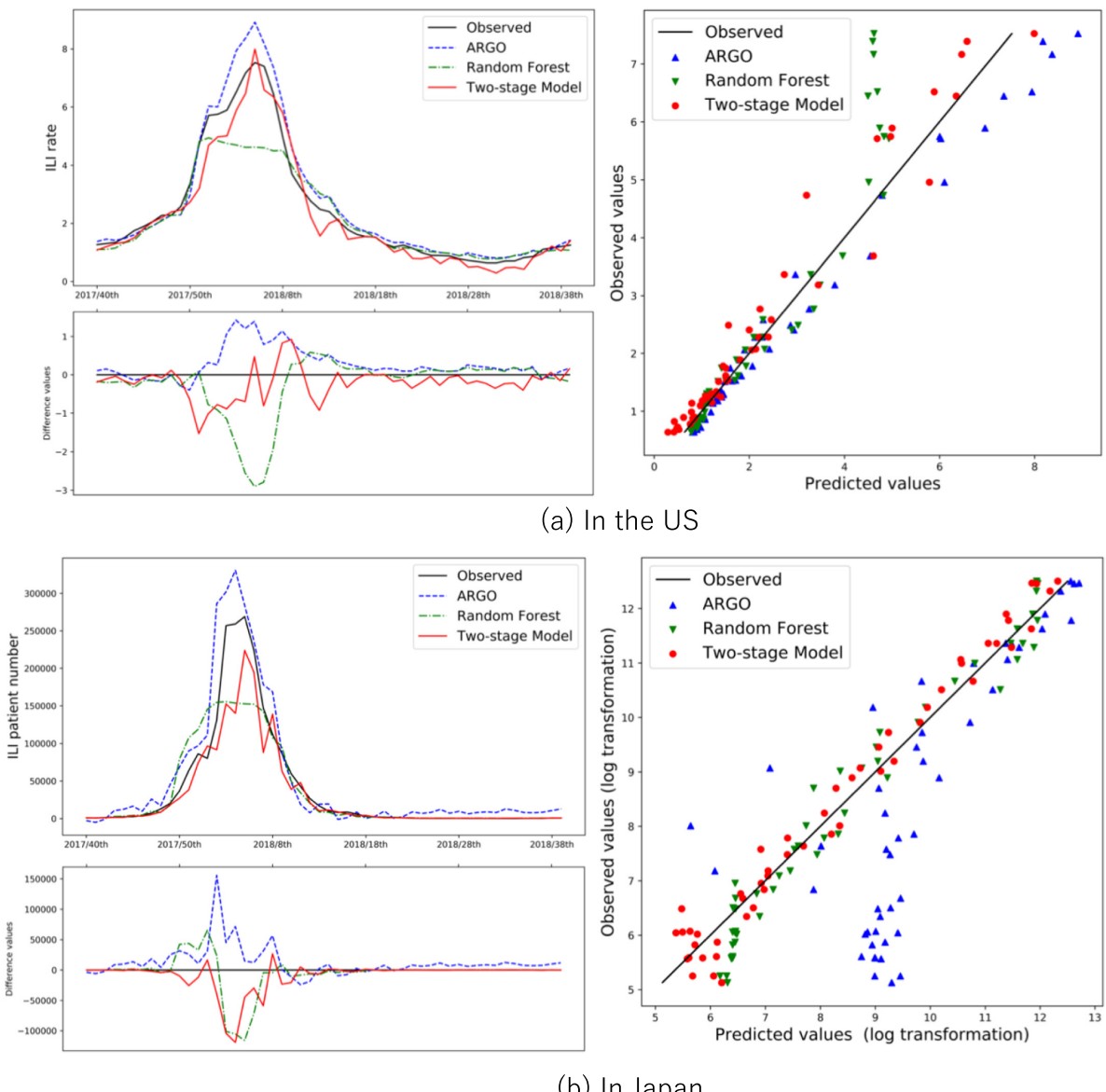

**Fig 4. The ILI rate and performance of the multiple models from the 2017/40th to the 2018/39th week in (a) the US and (b) Japan.** The left graph shows the time series (upper) and error plot (lower), which indicates the differences in between true and predicted ILI values of each model. The right graph shows observed vs. predicted scatter plot for each model.

Fig 4 shows the time series and observed (obs.) vs. predicted scatter plots of each model and the observed values in the US and Japan cases. The time series graphs show that the Two-stage model output predicted values close to the observed values. The model performed better at the beginning of epidemics than the other models in the case of Japan. The obs. vs. predicted scatter plots illustrate the prediction result of each model by period and country. It shows that ARGO tended to deviate upward while the Two-stage model and random forest tended to deviate downward. We need a longer observation period for the interpretation of the phenomena. The verification of how each model works for flu prediction is an issue for future studies.

**Table 3. Accuracy of each model on noisy data from the 2017/40th week to the 2018/39th week.** The values given in parentheses are the difference of accuracy scores between the original data and processed data with outliers.

| Outlier Ratio | Metrics | US | | | Japan | | |
|---|---|---|---|---|---|---|---|
| | | ARGO | RFR | Two-stage Model | ARGO | RFR | Two-stage Model |
| 0% | $R^2$ | 0.941 | 0.834 | 0.947 | 0.837 | 0.739 | 0.863 |
| | MAE | 0.329 | 0.432 | 0.321 | 16049.81 | 12143.82 | 10958.8 |
| | MAPE | 13.92 | 16.68 | 15.09 | 989.64 | 58.13 | 31.66 |
| 5% | $R^2$ | 0.917 | 0.801 | 0.934 | 0.845 | 0.645 | 0.836 |
| | | (-0.024) | (-0.033) | **(-0.013)** | **(+0.008)** | (-0.094) | (-0.027) |
| | MAE | 0.346 | 0.467 | 0.341 | 20861.21 | 16795.37 | 11923.57 |
| | | **(-0.017)** | (-0.035) | (-0.020) | (-4811.40) | (-4651.55) | **(-964.71)** |
| | MAPE | 15.14 | 15.70 | 13.79 | 1590.54 | 82.39 | 33.80 |
| | | (-1.22) | (+0.98) | **(+1.30)** | (-600.90) | (-24.26) | **(-2.14)** |
| 10% | $R^2$ | 0.912 | 0.756 | 0.937 | 0.815 | 0.439 | 0.760 |
| | | (-0.029) | (-0.078) | **(-0.010)** | (-0.022) | (-0.300) | (-0.076) |
| | MAE | 0.413 | 0.547 | 0.357 | 22813.21 | 21699.12 | 14242.20 |
| | | (-0.084) | (-0.115) | **(-0.036)** | (-6763.4) | (-5649.31) | **(-1807.61)** |
| | MAPE | 16.76 | 17.99 | 16.46 | 2534.14 | 107.69 | 36.85 |
| | | (-2.44) | **(-1.31)** | (-1.37) | (-1544.50) | (-49.56) | **(-5.19)** |
| 15% | $R^2$ | 0.860 | 0.722 | 0.927 | 0.735 | 0.167 | 0.777 |
| | | (-0.081) | (-0.112) | **(-0.020)** | (-0.102) | (-0.478) | **(-0.086)** |
| | MAE | 0.538 | 0.579 | 0.382 | 23000.91 | 27653.42 | 13879.09 |
| | | (-0.209) | (-0.147) | **(-0.061)** | (-6951.10) | (-11603.61) | **(-2170.72)** |
| | MAPE | 21.92 | 17.31 | 18.42 | 2610.99 | 113.94 | 39.61 |
| | | (-8.00) | **(-0.62)** | (-3.33) | (-1621.35) | (-55.81) | **(-7.95)** |

## Robustness

Table 3 presents the results of testing for accuracy and the differences in accuracy scores between the original data and processed data with outliers. The smaller the difference value is, the more robust the model is. In $R^2$, the 15% outliers reduced the performance by only -0.020 (compared with 0.112 for RFR, and 0.081 for ARGO), such a smaller difference against outliers shows the high robustness of the proposed model. Although under specified situations, the ARGO model in Japan for $R^2$ and random forest in the US for MAPE show better performance, the Two-stage model is relatively stronger. For example, the Two-stage model showed that the difference in MAPE is between +1.30 and -3.33 in the US and between -2.14 and -7.95 in Japan. Consequently, the results demonstrate that the Two-stage model is robust against outliers.

## Strengths and weaknesses of the two-stage model

Throughout the experiments, we demonstrate that historical ILI data and UGC data are useful for flu prediction, which is consistent with the results of previous studies [3, 6, 18]. One salient feature of our model is that the time series data spans are divided clearly into regular trend parts and irregular trend parts. We show that the proposed model, which processes the historical data separately from the UGC data as input, achieves better performance than the existing models that use these two input data simultaneously. Although this study employed the AR model as the 1st model, we could replace this part with any other models such as RNN [11] and disease model [31].

The proposed method has an advantage not only in performance, but also in robustness against outlier data. Noise and outliers included in UGC data, which are sometimes caused by transaction delays under heavy traffic or malfunctions in crawling, always hinder epidemic prediction tasks. It is crucial to build a model which is robust against outlier data for prediction, but most researches, except [8] and ours, have ignored the issue.

However, the proposed model has some limitations. Our approach depends strongly on the historical model produced by the 1st model. In flu prediction, the 1st model itself performs very well, which is suitable for our approach. However, in other targets such as infectious gastroenteritis (mainly norovirus) and Zika fever, which do not show any seasonal trends, the model mostly relying only on UGC data might perform better [18]. In future studies, a large scale comparative investigation that covers more infectious diseases would be desired.

## Conclusions

This study proposed a novel method for flu prediction: The Two-stage model that predicts regular trends from historical data and irregular trends from the UGC data. In the experiments conducted using the datasets of the US and Japan, we demonstrated that the proposed model can predict the ILI rate and the number of ILI patients with higher accuracy than existing models in the respective countries. The present results suggest that the proposed model is the most suitable for seasonal flu prediction among the compared models and that it is robust to outliers. Our model is only applicable to countries or regions where the amount of historical data is sufficient, which is the resource of the 1st model. Future studies must be conducted to develop a predictive model that can achieve higher accuracy without historical data or with fewer historical health reports.

## Supporting information

**S1 Table. Queries in English.**
(DOCX)

**S2 Table. Queries in Japanese.**
(DOCX)

## Author Contributions

**Conceptualization:** Taichi Murayama.

**Methodology:** Taichi Murayama.

**Supervision:** Eiji Aramaki.

**Writing – original draft:** Taichi Murayama.

**Writing – review & editing:** Nobuyuki Shimizu, Sumio Fujita, Shoko Wakamiya, Eiji Aramaki.

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
