## [Decision Letter · Decision Letter 0]

31 Jan 2020

PONE-D-19-34966

Robust Two-stage Influenza Prediction Model considering Regular and Irregular Trends

PLOS ONE

Dear Mr. Murayama,

Thank you for submitting your manuscript to PLOS ONE. After careful consideration, we feel that it has merit but does not fully meet PLOS ONE’s publication criteria as it currently stands. Therefore, we invite you to submit a revised version of the manuscript that addresses the points raised during the review process.

We would appreciate receiving your revised manuscript by Mar 16 2020 11:59PM. To enhance the reproducibility of your results, we recommend that if applicable you deposit your laboratory protocols in protocols.io, where a protocol can be assigned its own identifier (DOI) such that it can be cited independently in the future. For instructions see: http://journals.plos.org/plosone/s/submission-guidelines#loc-laboratory-protocols

We look forward to receiving your revised manuscript.

Kind regards,

Tzai-Hung Wen, Ph.D.

Academic Editor

PLOS ONE

Journal Requirements:

2. In your Methods section, please include additional information about your dataset and ensure that you have included a statement specifying whether the collection method complied with the terms and conditions for the website.

"This study was supported in part by JSPS KAKENHI Grant Number JP19K20279, Health and Labor Sciences Research Grant Number H30-shinkougyousei-shitei-004, and Yahoo! Japan.

We note that one or more of the authors is affiliated with the funding organization, indicating the funder may have had some role in the design, data collection, analysis or preparation of your manuscript for publication; in other words, the funder played an indirect role through the participation of the co-authors. If the funding organization did not play a role in the study design, data collection and analysis, decision to publish, or preparation of the manuscript and only provided financial support in the form of authors' salaries and/or research materials, please do the following:

Review your statements relating to the author contributions, and ensure you have specifically and accurately indicated the role(s) that these authors had in your study. These amendments should be made in the online form.Confirm in your cover letter that you agree with the following statement, and we will change the online submission form on your behalf:

"This study was supported in part by JSPS KAKENHI Grant Number JP19K20279, Health and Labor Sciences Research Grant Number H30-shinkougyousei-shitei-004, and Yahoo! Japan.

We note that you received funding from a commercial source: Yahoo Japan Corporation

Reviewers' comments:

Reviewer's Responses to Questions

5. Review Comments to the Author

Reviewer #1: The manuscript entitled “Robust Two-stage Influenza Prediction Model considering Regular and Irregular Trends” attempt to integrate time-series analysis and machine learning approaches to capture both regular trend and irregular patterns of influenza outbreaks. This is a well-written article and the result shows a promising direction for outbreak prediction of influenza.

Below are major issues

1. Page 5, Lines 131. Is there any reason to support the selection of AR(26)?

2. Figure 4. When comparing the difference values of each model, ARGO demonstrated a different direction of the deviance comparing to two-stage model and Random Forest approach. Is there any interpretation for the phenomena?

3. The result and discussion mainly focus on the performance comparison of different models. The content of discussion seems missed in this part. It’s better to add more detail discussion with previous literature in the manuscript.

Reviewer #2: The manuscript entitled ”Robust Two-stage influenza Prediction Model considering Regular and Irregular Trends” constructed a new Two-stage model to make real-time influenza forecasts based on the surveillance data and the google trend data in US and Japan. The manuscript is well organized, and the purpose of the work is clearly defined. Although the rationale of the approach presented is generally understandable, some points should be further clarified and described well.

My major concern is about the methods. Methods section lacks enough description of the main procedures of the Two-stage model. Very few words were used to describe the model. Author said “We designate the combination of the 1st Model and the 2nd Model as the Two-stage Model”, but how were these 2 models combined isn’t described. More details are needed to illustrate how to combine the two model’s results to obtain the final prediction value. Does the proposed approach only add the two models’ results? Or with any weights?

Also, how model 2 works with model 1’s outcome is not described. Eq(2) is not enough to understand how ILI rate from model 1 outcome can be used for model 2. Although “xt and ot-1 denote a current input and previous output vectors…” is described, what actual data are x and o, and how these variables correspond to ILI rate results from model 1.

Other minor comments are

- for Introduction

The introduction section should look like a concise summary of the research topic. I suggest the authors could show some previous studies' outstanding achievements based on the hospital surveillance data and UGC data (e.g. the quantity comparison of forecasting results based on the historical data and UGC data) rather than only explain the datasets.

On Page2, Ln 19, cite some particle filtering approaches done by Yang W and Shaman J for influenza forecasting.

On page 2, Ln 42, “We also think that our model is strong against noise and outliers in data”. Please change the work “think” to other words and describe your analytical results to support your statement. People think is not enough to support this conclusion.

- for Methods

What are the search queries used? Letting readers know these, e.g., by including it in supplementary materials, would be good.

Additionally, the merits of choosing ILI rate for US data and ILI patients for JPN data is not very clear to me.

Some variables in eq(2) are not expounded and need to be so.

- for Results

The mechanism shown as Figure 2 should be described in more details. I assume the solid black line represent the previous historical data. Then author should describe all the remaining dots (2 blue dots, one black, one red) following the order of the steps. For example, does left blue dot (prediction from the 1st model) occur before the above black dot. How is the 2nd blue dot predicted? And how is the red dot predicted? Why there is not red dot above the 1st left blue dot? Please describe the scenario more.

In Table 1, why the periods in training using Model1 and Model2 are different. Should both models use the same periods for training?

- for Robustness

The values given in parentheses are the difference of accuracy scores between the original data and processed data with outliers processed data. Please describe more details what is processed data and how did author process the data.

- for Conclusions

- on the sentence "The model presented herein might not work well in some countries": if the authors decide to include this sentence, they should preferably give clear reasons why and how the model can be made to work in such countries.

6. PLOS authors have the option to publish the peer review history of their article (what does this mean?). If published, this will include your full peer review and any attached files.

Reviewer #1: No

Reviewer #2: No

---

## [Author Response · Author response to Decision Letter 0]

18 Mar 2020

To the reviewr and editors:

Please check our Response to Reviewers file (rebuttal letter), in which we described responses to the editor and review's comments in detail.

Add the detailed comments about backing revisions to cover letter.

---

## [Decision Letter · Decision Letter 1]

29 Apr 2020

Robust Two-stage Influenza Prediction Model considering Regular and Irregular Trends

PONE-D-19-34966R1

Dear Dr. Murayama,

We are pleased to inform you that your manuscript has been judged scientifically suitable for publication and will be formally accepted for publication once it complies with all outstanding technical requirements.

With kind regards,

Tzai-Hung Wen, Ph.D.

Academic Editor

PLOS ONE

Additional Editor Comments (optional):

Reviewers' comments:

Reviewer's Responses to Questions

**Comments to the Author**

1. If the authors have adequately addressed your comments raised in a previous round of review and you feel that this manuscript is now acceptable for publication, you may indicate that here to bypass the “Comments to the Author” section, enter your conflict of interest statement in the “Confidential to Editor” section, and submit your "Accept" recommendation.

Reviewer #1: All comments have been addressed

Reviewer #2: All comments have been addressed

2. Is the manuscript technically sound, and do the data support the conclusions?

Reviewer #1: Yes

Reviewer #2: Yes

3. Has the statistical analysis been performed appropriately and rigorously? 

Reviewer #1: Yes

Reviewer #2: Yes

4. Have the authors made all data underlying the findings in their manuscript fully available?

Reviewer #1: Yes

Reviewer #2: Yes

5. Is the manuscript presented in an intelligible fashion and written in standard English?

Reviewer #1: Yes

Reviewer #2: Yes

6. Review Comments to the Author

Reviewer #1: (No Response)

Reviewer #2: (No Response)

7. PLOS authors have the option to publish the peer review history of their article (what does this mean?). If published, this will include your full peer review and any attached files.

Reviewer #1: No

Reviewer #2: Yes: Hsiang-Yu Yuan

---

## [Editor Report · Acceptance letter]

5 May 2020

PONE-D-19-34966R1 

Robust Two-stage Influenza Prediction Model considering Regular and Irregular Trends 

Dear Dr. Murayama:

I am pleased to inform you that your manuscript has been deemed suitable for publication in PLOS ONE. Congratulations! Your manuscript is now with our production department. 

With kind regards,

on behalf of

Dr. Tzai-Hung Wen 

Academic Editor

PLOS ONE